# Needle Tip Detection Using Ultrasound Probe for Vertical Punctures: A Simulation and Experimental Study

**DOI:** 10.3390/diagnostics12020527

**Published:** 2022-02-18

**Authors:** Yuusuke Tanaka, Katsuhiko Tanaka, Hisanori Shiomi, Yoshimasa Kurumi, Tohru Tani, Yukio Ogura

**Affiliations:** 1Japan Probe Co., Ltd., 1-1-14 Nakamura-Cho, Minami-ku, Yokohama-shi 232-0033, Kanagawa, Japan; ogura@jp-probe.com; 2Research Organization of Science and Technology, Ritsumeikan University, 1-1-1, Nojihigashi, Kusatsu-shi 525-8577, Shiga, Japan; tanaka-k@fc.ritsumei.ac.jp; 3Department of Surgery, Nagahama Red Cross Hospital, 14-7, Miyamae-cho, Nagahama-shi 526-8585, Shiga, Japan; shiomi@belle.shiga-med.ac.jp; 4Japan Community Health Care Organization Shiga Hospital, 16-1, Fujimidai, Otsu-shi 520-0846, Shiga, Japan; kurumi@belle.shiga-med.ac.jp; 5Department of Research and Development for Innovative Medical Devices and Systems, Faculty of Medicine, Shiga University of Medical Science, Tsukinowa-cho, Seta, Otsu-shi 520-2192, Shiga, Japan; tan@belle.shiga-med.ac.jp

**Keywords:** puncture, ultrasound waves, ultrasound-guided, needle tip detection, guided wave

## Abstract

Current ultrasound-guided punctures are difficult to perform as they are performed at an angle to the ultrasound image of the affected area, resulting in longer puncture times, lower success rates, and higher unexpected injury rates. Vertical puncture techniques have also been investigated, but the principle of needle tip detection remains unclear. To optimize ultrasound probes for puncture, the principle of needle tip detection should be understood. This study aimed to verify the principle of needle tip detection and optimal measurement conditions for vertical puncture. Needle tip detection was performed in animal experiments using a probe with a central puncture slit. Moreover, the needle tip was detected at short distances using a puncture spacer. We also investigated the signal from the needle tip using a ring probe and confirmed the principle of needle tip detection, effect of needle tip angle, and insertion depth on needle tip detection through simulation and experiments. Needle tip detection using ultrasound-guided waves was described, and the relationship among needle tip angle, detection intensity, and phase change was verified. The needle tip can be detected by the leakage of the ultrasound-guided wave generated inside the needle tip.

## 1. Introduction

In the medical field, ultrasound-guided puncture of an affected area is performed while observing the position of the affected area and needle. In Berlyne’s study, renal biopsy was performed using A-mode-guided ultrasound [1]. In this method, the change in the thickness of the kidney was measured during synchronization with breathing, but the needle was not detected simultaneously. A method for needle detection during puncture using ultrasound waves was reported by Kratochwil et al. [2], Goldberg et al. [3], Holm et al. [4], and Itoh et al. [5]. In their studies, an ultrasound probe with a through-hole in the center was used for the puncture while monitoring the needle tip using A-mode images. Amniotic fluid puncture, body fluid aspiration, and puncture of the liver and pancreas were reported. Rasmussen et al. [6] and Kristensen et al. [7] performed liver and kidney biopsies by ultrasonically guided vertical puncture using an ultrasound probe with a through-hole. These punctures were made while detecting both the affected area and needle tip and were easier than blind punctures, such as that in the Menghini method. These studies did not mention the principle of needle tip detection in vertical puncture.

Puncture has been performed while detecting the needle on the B-mode image, using a linear array ultrasound probe with a puncture attachment [8,9], a linear array probe with a through-hole [10], or a linear array probe with a V-shaped groove at the center [11]. The puncture attachment has been reported for use with a sector scanning ultrasound probe [12].

In endoscopic ultrasound-guided fine needle aspiration (EUS-FNA), the affected area is punctured obliquely using a dedicated ultrasound probe. The oblique puncturing method requires considerable skills. Puncture by forward-view EUS-FNA has also been proposed, which allows ease of puncture [13] and superior success rate and treatment time [14]. The puncture is performed in the same direction as the progression of the endoscope during forward-view EUS-FNA [15], but the ultrasound B-mode image is inclined 45° with respect to the advancing direction of the endoscope. The puncture needle is inserted from an oblique direction on the B-mode image [13]. Therefore, operability improves by matching the directions of the probe, ultrasound propagation, and puncture. Figure 1 shows the direction of the probe, transmission direction of the ultrasound wave, insertion direction of the puncture needle, and region of ultrasound emission or imaging. All directions are similar in vertical puncture (Figure 1a). In the puncture attachment (Figure 1b), the direction of the probe and the transmission direction of the ultrasound wave do not coincide with the direction of the puncture. In oblique-view EUS-FNA (Figure 1c), no directions coincide, and in forward-view EUS-FNA (Figure 1d), the directions of ultrasound transmission and puncture are different. Moreover, the vertical puncture shown in Figure 1a provides the best operability during puncture because all directions coincide.

The needle for puncture has various shapes to facilitate detection; the uneven surface of the needle combined with phased array may allow needle tip detection [16,17]. One study that compared core biopsy and vacuum-assisted breast biopsy in puncture emphasized the lightness of the probe and intuition of puncture [18]. Additionally, especially processed needles and external devices have high costs. Therefore, intuitive puncture without using a complicated device is necessary, and vertical puncture could satisfy this condition.

Therefore, vertical puncture decreases the puncture time and increases the success rate by intuitive operation. The principle of needle tip detection for optimizing vertical puncture should be elucidated. This study aimed to verify needle tip detection and optimum measurement conditions in vertical puncture. We have detected the needle tip with forward-view ultrasound waves using a ring-type array probe [19] and explained the principle of needle tip detection with edge waves [20]. Furthermore, we punctured a phantom and porcine liver and detected the needle tip in real time during forward view [21,22]. Details on ultrasound and edge waves generated from the probe have been previously reported [23]. However, the phenomena cannot be explained by the edge waves as the relationship between needle tip angle and received signal intensity. The effect of the probe slit on needle tip detection and methods for detecting the needle tip at short distances have not been evaluated [21]. This study aimed to elucidate the principle of needle tip detection, investigate the effect of the probe slit on needle tip detection, and develop a method for detecting needle tips at short distances. The relationship between the needle tip angle and received signal intensity was investigated, and the edge wave cannot explain the change in signal intensity. In the edge wave, although the ultrasound propagation phenomenon is observed only in water [20], from the simulation where the material of the needle compared is stainless steel and void, ultrasound propagation is noted near the needle tip. We assumed that ultrasound propagation at the needle tip is the same phenomenon as the guided waves generated by a thin plate and examined the changes in received signal intensity and phase by the simulation. The detection of the needle tip at short distances has been solved by developing a puncture spacer and attaching it to the probe.

## 2. Materials and Methods

### 2.1. Animal Statement

All animal experiments were conducted with the approval of the Animal Use Committee of Shiga University of Medical Science, Shiga, Japan (approval number, 2013-6-8). An adult female pig (weight, 40 kg) and two female beagles (weight, 8 kg) purchased from Shimizu Laboratory Supplies Co., Ltd. (Kyoto, Japan) were used in this study. The animals were bred under specific pathogen-free conditions and cared for following the guidelines of the Guide for the Care and Use of Laboratory Animals (National Institutes of Health, Bethesda, MD, USA) [24] and animal research guidelines of Shiga University of Medical Science. After the intramuscular administration of ketamine hydrochloride (10 mg/kg), anesthesia was maintained with inhalation of 2% isoflurane with spontaneous ventilation during surgery. Saline (0.9%) was administered during anesthesia. The pig was sacrificed under anesthesia after liver removal.

### 2.2. Detection of the Needle Tip in the Phantom and Animal Experiments

#### 2.2.1. Ultrasound Probes for Vertical Puncture

We previously developed a prototype ultrasound probe for vertical puncture with needle tip detection (Appendix A) and performed punctures with needle tip detection in phantoms and porcine liver [20,21]. Previously, the ring probe was used to detect the needle tip, but the probe had to be removed while the puncture needle was still in place. To remove the probe with the needle in place, we developed a prototype ultrasound probe with a U-shaped slit in the center (Appendix A). A U-shaped ultrasonic transducer (outer and inner diameters, 7 and 3 mm, respectively) was placed, surrounded by 11 fan-shaped transducers (outer and inner diameters, 17.8 and 8 mm, respectively). The slit was placed to detach the probe while keeping the puncture needle close to the affected area.

#### 2.2.2. Needle Tip Detection in Porcine Liver

The removed porcine liver was punctured as a preliminary experiment before an experiment on the living tissue to verify needle tip detection via vertical puncture. We used the U-shaped slit ring-type array probe (Appendix A). The porcine liver was placed in water and measured by applying a pulse voltage of −48 V and 7 MHz from a pulser-receiver (Krautkramer Japan, Tokyo, Japan, PAL3), and a B-mode image was obtained. A needle with a tip angle of 18° (Terumo, Tokyo, Japan, Surflo indwelling needle) and diameter of 0.95 mm was inserted through the through-hole, and the porcine liver was punctured.

#### 2.2.3. Needle Tip Detection in Canine Femoral Artery

The femoral artery of a dog was punctured to detect the needle tip in the living tissue using a linear array probe (Appendix A). A pulsed voltage of −48 V and 7 MHz was applied to the linear array probe from a pulser-receiver (Krautkramer Japan [now KJTD], Tokyo, Japan, PAL3) to measure the B-mode image. Detection of a needle at a short distance via a vertical puncture is difficult, as reported in a previous study [21]. Therefore, to detect the needle tip at a short distance, a puncture spacer (Figure 2) was used (Figure 3). The puncture was performed (Figure 3) with the puncture spacer attached. A stainless steel needle with a diameter of 0.95 mm and tip angle of 18° (Terumo, Tokyo, Japan, Surflo indwelling needle) was used to puncture the femoral artery of the dog from the U-shaped slit.

### 2.3. Principle of Needle Tip Detection

#### 2.3.1. Basic Model Simulation

The detection of the tip of a stainless steel needle placed in water has been verified by simulation [21]. Moreover, the phase change was confirmed experimentally, and the detection principle was explained by the edge wave [20]. However, since ultrasound propagation was considered only in water, the phenomenon of ultrasound propagation inside the needle was not considered, and the relationship between the needle tip angle and detection intensity described below could not be explained. In this study, we simulated a needle in water in the case of stainless steel and void to verify the propagation of ultrasound inside the needle. The received waveform was confirmed via ultrasound wave propagation simulation to elucidate the detection principle of needle tip position. An ultrasound propagation simulator (eCompute Corporation, Ehime, Japan, SWAN 21) was used with an elastodynamic finite integration technique. A two-dimensional model was used (Appendix A): a needle was inserted into a ring-type transducer with outer and inner diameters of 7 and 3 mm, respectively; the medium was water, the needle material was stainless steel (SUS 304), the diameter was 1.5 mm, and the tip angle was 10°. The tip angle and needle insertion depth were set to 10° and 15 mm, respectively, because sensitivity was high in the experiments described later. The boundary conditions of the left and right sides and lower end of the model were set as the absorption boundary. The ultrasound wave was generated from element A and received by elements A and B; then, the waveforms at elements A and B were compared. The physical properties of the materials were as follows. In the water, density was set to 1000 kg/m^3^, and acoustic velocity was set to 1500 m/s. In the SUS 304 of the needle (Appendix A), density was set to 7930 kg/m^3^, longitudinal wave acoustic velocity was set to 5780 m/s, and shear wave acoustic velocity was set to 3100 m/s. The needle is void (Appendix A). The transmission waveform is expressed using the following Equation (1):(1)$$y=-\cos\left(2\pi ft\right)\left(1-\cos\left(1-\frac{2\pi ft}{n}\right)\right)$$
where *y* is the input surface force, *f* is the frequency, *t* is the time, and *n* is the wave cycle.

The frequency was 7 MHz, the wave cycle was 2 and surface force was applied to element A.

Next, tip detection on the cylindrical model shown in Appendix A was verified using simulation. In the stainless steel (SUS 304) and void cylinder models shown in Appendix A, respectively, a negative pulse was applied to the A side under the same conditions as before and received by elements A and B. The material properties are similar as those in Appendix A.

To investigate the propagation of ultrasound waves inside the needle, ultrasound propagation was evaluated in the model shown in Appendix A. A 3.3-mm-wide ultrasonic probe was placed on a 0.2-mm-wide stainless steel plate in the water at a 36° angle. Waveform output points were placed at points A and B and waveforms at each position were examined. This model aimed to check the generation of guided waves. Guided waves are generated when ultrasound waves are incident on a thin plate, and the longitudinal and transverse waves are combined to generate ultrasound waves of large amplitude. We assumed that the guided waves were generated near the needle tip, which can be approximated as a thin plate, and examined whether the guided waves were generated at 7 MHz, which is the transmission frequency.

#### 2.3.2. Angle of Needle Tip Detection Sensitivity

The relationship between tip angle and reception strength has been investigated experimentally [20] but was not compared with simulations. Moreover, we have not measured the orientation of the needle to match the simulation. The relationship between the angle of the needle tip and detection sensitivity was investigated. Six types of stainless steel (SUS 304) needles with a diameter of 1.5 mm were prepared, with tip angles of 5°, 10°, 15°, 20°, 30°, and 40°; these were used to measure the signal intensity from the needle tip. The separated ring-type probe shown in Appendix A was used. The needle tip was inserted to a depth of 15 mm from the front of the probe, and a negative spike pulse of −68 V was applied to element A using a pulser-receiver (Imaging Supersonic Laboratories, Nara, Japan, KitPRAD2d). The probe was placed in water such that the tip position of the needle was 15 mm deep from the probe surface. The slope of the inserted needle was oriented to elements A as shown in Appendix A, and a negative pulse of −68 V was applied to element A and received by elements A and B.

Next, the relationship between the needle tip angle and reception strength was investigated using a simulator (eCompute Corporation, Ehime, Japan, SWAN 21). The simulation model is shown in Appendix A, and the angle of the needle tip was changed from 1° to 40°. The ultrasound wave was generated from element A and received by elements A and B, and the signal intensity at elements A and B was evaluated. The material constants and transmission waveforms are the same as those in the previous section.

#### 2.3.3. Needle Tip Width and Length Modification Model

To validate the needle tip detection using guided waves, it is necessary to study the ultrasound propagation at different widths and lengths of the tip.

The principle of needle tip detection was verified using the simulation model shown in Appendix A. The tip width of the two cases in the simulation model changed from 1 to 10 mm, and the tip length was 5 mm (Appendix A). The tip width was 0.2 mm, and the tip length *l* changed from 0.1 to 1.0 mm (Appendix A). The ultrasound wave was generated from element A and received by element A; next, the intensity of the received waveform at element A was evaluated. The material constants and transmission waveforms are similar to those in the previous section.

## 3. Results

### 3.1. Needle Tip Detection at the Phantom and Animal Experiments

#### 3.1.1. Needle Tip Detection in Porcine Liver

We used type A (Appendix A) and type B (Appendix A) ring array probes to detect needle tips in the vascular phantoms [21] and porcine liver [20]. It is necessary to investigate the effect of the slit of type C (Appendix A) on needle tip detection. Type B without the slit detects the needle tip, as shown in Appendix A. Figure 4 shows the B-mode image with the U-shaped slit ring-type array probe (type C). Figure 4a shows the image of the needle tip near the liver surface, and Figure 4b shows the image when the needle tip was near the bottom surface of the liver. As with the Type B probe, the needle tip position could be detected in real time. The slit part did not affect the needle tip detection.

#### 3.1.2. Needle Tip Detection in the Canine Femoral Artery

When a type D probe (Appendix A) was used to puncture a dog, it was difficult to detect the needle tip because of noise near the surface [21]. To detect the needle tip near the surface, the results of puncture with a probe equipped with a puncture spacer are described. Figure 5 shows an image of puncturing a canine femoral artery using a puncture spacer (Figure 2). Arteries, needle tips, and body surfaces were detected in the dog, which confirmed that the needle tip can be detected using spacers, even in the vicinity of a living body surface.

### 3.2. Experiments of Needle Tip Detection

#### 3.2.1. Simulation of the Basic Model

Simulations have shown ultrasound propagation at the needle tip [21], but the received waveform has not been verified. The ultrasonic wave propagation near the needle tip has also not been verified, whether it is a phenomenon in water or inside the needle. The received waveform and ultrasound propagation near the needle tip are described below. Figure 6 shows the ultrasound propagation obtained from the simulation. The direction of the receiving side surrounding the needle and signal of the direction returning to the transmitting side was detected. Appendix A shows the received waveforms of elements A and B. The phases of the received waveforms were opposite to each other. Therefore, the ultrasound wave of the receiving side surrounding the needle and that returning to the transmitting side are different signals.

Figure 7 shows the simulation results at the void needle tip. Only 1% of the signal intensity was detected from the needle tip of the gap compared to the stainless steel needle tip. An animation comparing Figure 6 and Figure 7 is shown in Appendix A.

Next, Appendix A shows the simulation results from the cylinder tip. The ultrasound wave detected at the stainless steel cylinder tip was 100 times larger than that at the void cylinder tip. Ultrasound waves were also detected in the voids but only weak signals were detected, similar to the needles. As in the case of needles, a video comparing stainless steel and void cylinders is shown in Appendix A. Thus, stainless steel affects the signal amplification of ultrasound waves. Appendix A shows the received waveforms of elements A and B in the stainless steel cylinder. The phases of the maximum amplitude signals were reversed from each other. Moreover, the time difference between the positions of the peak signals p and n became 1 μs. Based on these results, the ultrasound waves are entering the stainless steel at the needle tip and cylinder corners, and some type of ultrasound wave is generated at the needle tip or tip corner of the cylinder.

The simulation results for the model in Appendix A are shown in Appendix A. Guided waves were generated in the stainless steel plate and leaked into the water. White indicates positive amplitude, and black indicates negative amplitude. The received waveforms at points A and B in Appendix A are shown in Appendix A, where the phase of the incoming ultrasound wave (left) is opposite to that of the opposite wave (right).

#### 3.2.2. Angle of Needle Tip Detection Sensitivity

Figure 8 shows the results of the needle tip angle and received signal intensity at the experiment and simulation. The received intensity increased with needle tip angles of 10° and 15°. Because the tip angles of the injection needles used in the medical field are 12°, 15°, and 18°, large intensity values can be obtained.

#### 3.2.3. Needle Tip Width and Length Modification Model

This section describes the simulation results when the tip width *w* and tip length *l* in Appendix A are changed to determine the effect of guided waves on needle tip detection. Appendix A shows the reception amplitude of element A when changing tip width *w*. The tip width of 1.5 mm is similar to that of the cylinder. The amplitude increased when the tip width was 0.2 mm, but the value was nearly constant when the tip width was ≥ 0.5 mm. Appendix A shows the reception amplitude of element A when changing tip length *l*. The amplitude was maximized at a tip length of 3 mm. Appendix A present the ultrasound propagation diagrams with tip widths *w* of 0.2 mm and 0.5 mm, respectively. When *w* is 0.2 mm, a large ultrasound wave is generated inside the stainless steel cylinder, and ultrasound wave x leaks opposite the incident side (Appendix A). As shown in Appendix A, a large ultrasound wave is received by the reflection at the tip. Conversely, when *w* is 0.5 mm, the ultrasound wave generated inside the stainless steel cylinder is small (Appendix A), and the signal at the time of reflection is also small (Appendix A). Appendix A show the ultrasound propagation movies for different tip width *w* and tip length *l*. Similar to the ultrasonic propagation of the cylinder in Appendix A, ultrasonic waves were generated at the tip corner. Therefore, it is observed that an ultrasound-guided wave is generated in the portion where the width is reduced.

## 4. Discussion

In vertical punctures, the needle tip was detected in the porcine liver using a U-slit probe and ring probe. The needle tip was also detected in a canine femoral vessel near the body surface using a puncture spacer. Ultrasound was incident on the narrow part near the needle tip, and ultrasound-guided waves were generated. The effect of the slit on the detection of the needle tip is discussed: the needle tip was detected during the puncture of the porcine liver with the U-slit probe (Figure 4), similarly near the surface and back of the liver. Therefore, the slit has no effect on the detection of the needle tip in the vertical puncture. The detection of the needle tip near the body surface using the puncture spacer is discussed. In the canine thigh puncture, it has been difficult to detect the needle tip at a short distance near the body surface [21]. In vertical puncture, the received signal was small at short distances due to the large angle of incidence of the ultrasound from the needle tip to the ultrasound transducer. However, using a puncture spacer, the distance was increased, and the angle of incidence was reduced; thus, the needle tip could be detected even near the body surface. Considering the cost, vertical puncture does not require external devices, such as magnetic sensors and needle tip vibrators [25]. Moreover, Figure 8 shows that the needle tip angle of 10°–20° is sensitive enough to allow the needle tip to be detected with a normal needle without the need for needle modification [18]. Therefore, the cost of the puncture is expected to reduce.

Next, we consider the principle of needle tip detection. The principle of needle tip detection was explained by Goldberg in terms of the change in the cross-sectional area of the ultrasound beam [26], but there are some phenomena that cannot be explained, such as the change in phase. Although the principle of needle tip detection by edge waves has been explained in previous studies [21], the difference in received intensity depending on the angle and width of the needle tip has not been explained in previous studies. Furthermore, Appendix A shows that the ultrasound generated by the stainless steel needle tip is clearly larger than that generated by the voided needle tip. Furthermore, Appendix A shows the difference in behavior between the stainless steel cylinder tip and that of the voided cylinder. These results show that ultrasound is generated at the thin parts, such as the stainless steel needle tip and cylinder corners. The ultrasound from the tip of the void can be explained by edge waves, but other phenomena should be considered to explain the findings in stainless steel.

First, we consider the phase change in the ultrasound wave generated on the stainless steel plate, which is shown in Appendix A. In the case of the guided wave generated at the needle tip, the phase of the ultrasonic wave changes according to the direction of leakage; thus, the received signals on the A and B sides of Figure 6 are opposite in phase (Appendix A). Moreover, the phenomenon that the received signals of the A and B sides of the cylinder in Appendix A are out of phase (Appendix A) can be explained similarly. If we assume that the guided wave is generated at the cylinder corner and the needle tip, the guided wave is generated twice at the corner of the A and B sides. As shown in Appendix A, the guide wave generated at the corner of A side is opposite in phase to the guided wave 1 that leaked to element A side and guided wave that leaked into the water of the flat end of the cylinder tip side. Because the guided wave to element A is in the opposite phase to the transmission waveform, the guided wave to the cylinder tip is in the same phase as the transmission waveform. The guided wave that leaked to the cylinder tip generates another guided wave at the corner of the B side. At this time, the guide wave that leaked to the cylinder tip is in the opposite phase to the original guide wave, and the guide wave 2 that leaked to element B is in the same phase. Therefore, the phase of the signal received at element A and signals 1 and 2 received at element B are opposite. In Appendix A, the time difference between signal n and signal p is 1 μs, but the time for the ultrasonic wave to pass through the cylinder tip is 1 μs because the width of the cylinder tip is 1.5 mm and speed of sound in water is 1500 m/s. From these results, the phase change can be explained by the generation of a guided wave at the needle tip.

Next, we consider the relationship between the needle tip angle and reception intensity from the simulation results of Appendix A for different tip widths *w* and lengths *l*. The simulation results for different tip widths *w* in Appendix A show that the reception intensity is higher when the width is 0.2 mm, and the intensity of the guided wave is the highest. The simulation results for different tip lengths *l* in Appendix A show that the received intensity is higher when the tip length is 3 mm. These phenomena are explained in Appendix A. In Appendix A, the guided wave is generated at the tip, but the ultrasound wave *x* is always leaked. Therefore, if the tip is extremely long, the leakage signal will be large, and the reception strength will be small. Conversely, if the tip is extremely short, it is considered that there is inadequate space for the longitudinal and transverse waves to be combined in the stainless steel, and the received intensity becomes small. In this model, a tip length *l* of 3 mm provides a good balance between the generation of guided waves and leakage into the water. When the tip width *w* is increased (Appendix A), the guided wave is no longer generated, and two signals with time difference are generated at the tip. This is because the guided wave is generated at the corner of the cylinder (Appendix A). Therefore, the tip width was no longer affected, and for w ≥ 0.5, the guided wave at the tip corner was detected as in the phenomenon at the cylinder tip. These results suggest that the guided waves at the needle tip were largely generated at the tip where the thickness was 0.2 mm. The detection intensity of the guided wave is reduced if the propagation distance is extremely long or short. Therefore, it can be assumed that the tip angle between 10° and 20° is suitable for the propagation distance of the guided wave under the condition of 7-MHz frequency and stainless steel needle, and the amount of guided wave generation is large.

From these results, it can be explained that the needle tip can be detected by the guided waves generated near the tip. The guided wave is generated at the thinner part of the needle tip, the guided wave inside the stainless steel needle is reflected by the needle tip, and the reflected guided wave leaks into the water. When the thickness of the stainless steel plate is constant, the angle of leakage of the guided wave is constant, but at the needle tip, because of the change in thickness, the guided wave leaks at various angles. Therefore, it is considered that the guided wave leaked in a spherical shape at the needle tip.

However, this study has its limitations. The operability of vertical puncture has not been quantitatively tested. In EUS-FNA, anterior-view puncture is superior to oblique puncture in terms of success rate and treatment time [14], and the same effect can be expected for vertical puncture. In the future, we will examine the operability of vertical puncture based on the success rate and puncture time.

Some applications of vertical puncture are presented in Table 1. Previously, amniotic fluid puncture, aspiration of body fluids, and biopsy of the kidney have been performed [2,3,4,5,6,7,10], and vertical puncture can be used for various purposes. As for needle tip detection at the time of puncture, it is expected to be applied in epidural puncture and central venous catheter insertion and will be studied in the future.

In conclusion, it was confirmed that the needle tip can be detected by a normal needle with a tip angle of 12°–18° in the vertical puncture, the needle tip can be detected at a short distance into vertical puncture using a puncture spacer, the slit in the ultrasonic probe for puncture does not affect the needle tip detection, and the needle tip is detected by the guided wave generated inside the vicinity of the needle tip in the vertical puncture.

## 5. Conclusions

We have developed two types of ultrasound probes for vertical puncture: a separate ring ultrasound probe for checking the needle tip detection signal and four different array probes for image processing. We performed a puncture while observing the needle tip in real time with a pig liver and confirmed that the slit for removing the probe did not affect the needle tip detection. We made a prototype of a puncture spacer to detect the needle tip at a short distance and detected the needle tip in the canine femoral artery. The relationship between the needle tip angle and reception strength was checked, and the sensitivity of the needle tip detection became high at a tip angle of 10°–20°, confirming the vertical puncture using a normal needle. The principle of needle tip detection was explained by the guided wave generated near the needle tip. The use of an ordinary injection needle instead of a special needle with a modified tip makes it possible to perform vertical puncture at low cost. Having verified the principle of needle tip detection and shape of the needle, it will be possible to develop an ultrasound probe with a suitable shape for vertical puncture. In the future, we will also examine the success rate of vertical puncture and reduction in puncture time to reduce the number of puncture accidents.

## Figures and Tables

**Figure 1 diagnostics-12-00527-f001:**
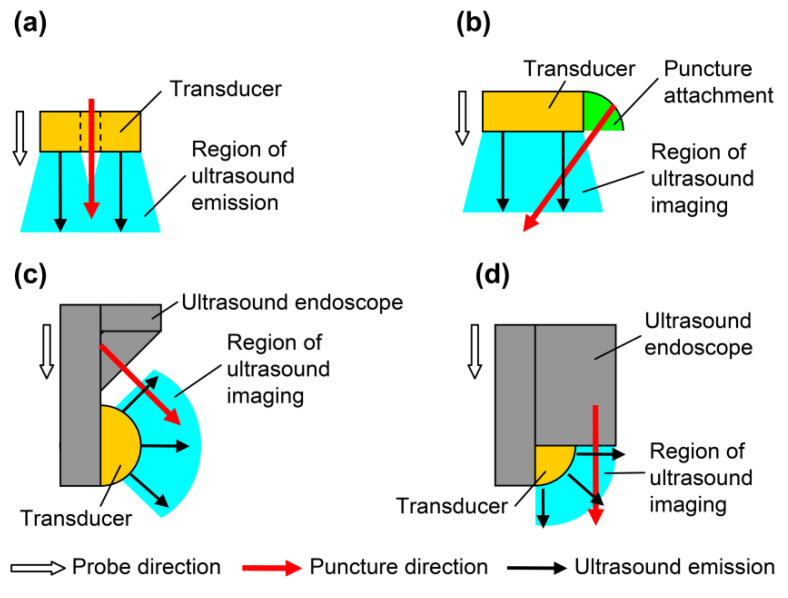
Directions of probes for puncture: probe direction, puncture direction, and region of ultrasound imaging. (**a**) Simple probe with through-hole; (**b**) linear array with puncture attachment; (**c**) oblique-view endoscopic ultrasound-guided fine needle aspiration (EUS-FNA); and (**d**) forward-view EUS-FNA.

**Figure 2 diagnostics-12-00527-f002:**
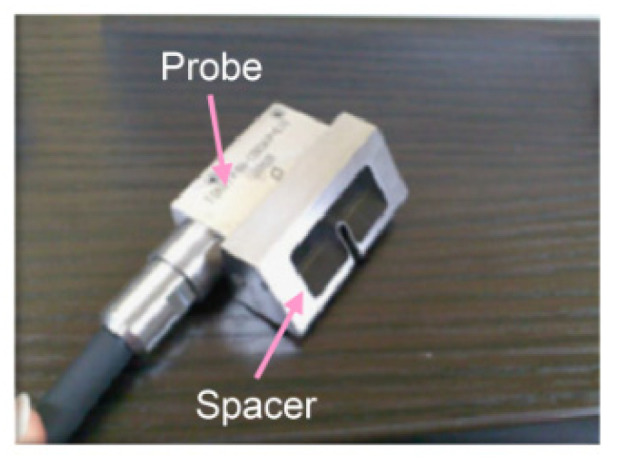
Equipped spacer for puncture [22]. Copyright(c) 2014 IEICE.

**Figure 3 diagnostics-12-00527-f003:**
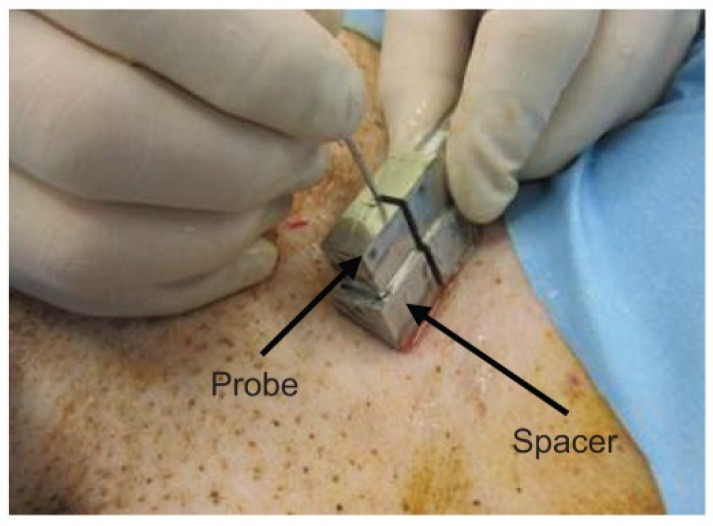
Puncture with the spacer [22]. Copyright (c) 2014 IEICE.

**Figure 4 diagnostics-12-00527-f004:**
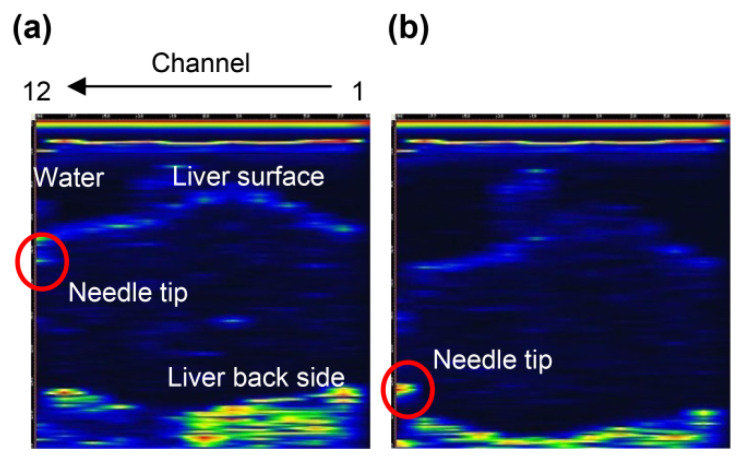
B-mode images of the needle tip and porcine liver using a U-slit ring-type array probe (type C): (**a**) needle tip near the surface of the liver and (**b**) needle tip near the back of the liver.

**Figure 5 diagnostics-12-00527-f005:**
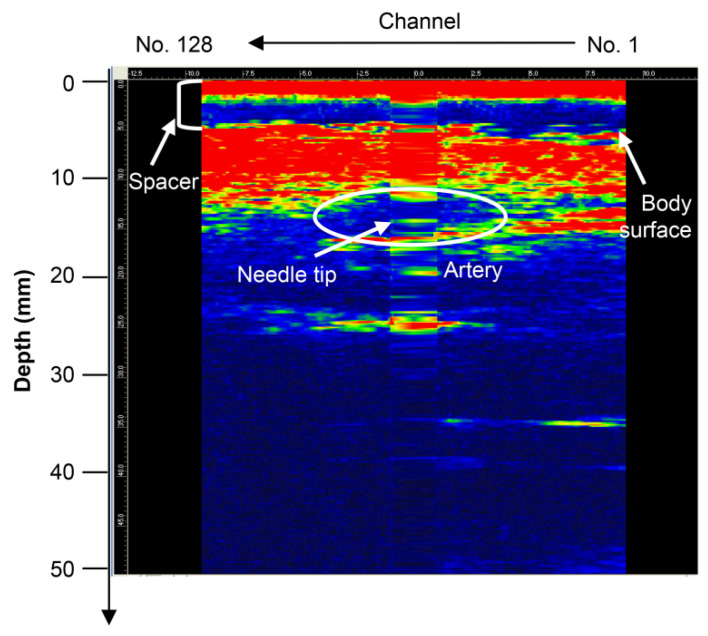
Puncture of the canine femoral artery using the type D probe (with puncture spacer) [22]. Copyright (c) 2014 IEICE.

**Figure 6 diagnostics-12-00527-f006:**
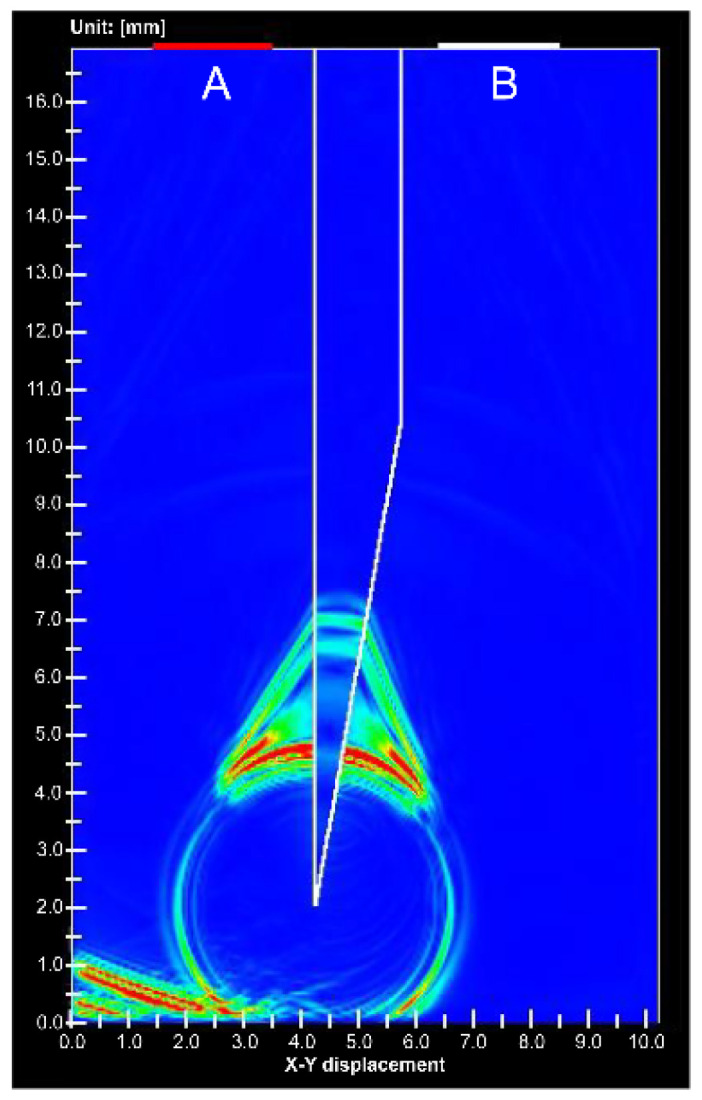
Simulation image of ultrasound propagation at the stainless needle tip. (**A**) is ultrasound transducer for transmitting and receiving, (**B**) is receiving transducer.

**Figure 7 diagnostics-12-00527-f007:**
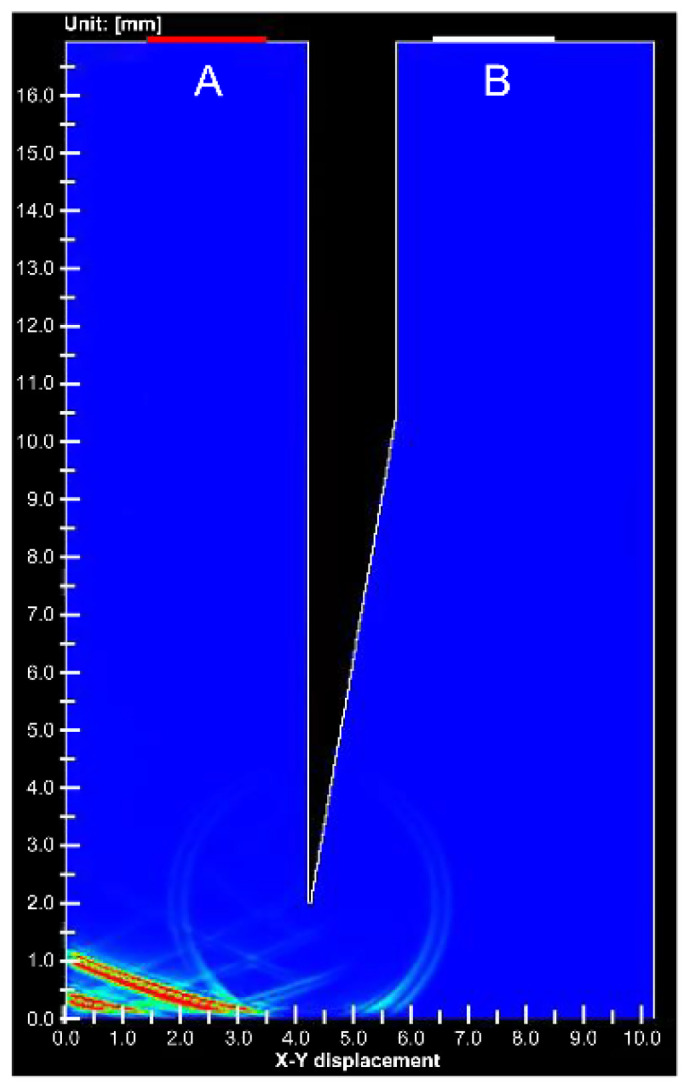
Simulation image of ultrasound propagation at the void needle tip. (**A**) is ultrasound transducer for transmitting and receiving, (**B**) is receiving transducer.

**Figure 8 diagnostics-12-00527-f008:**
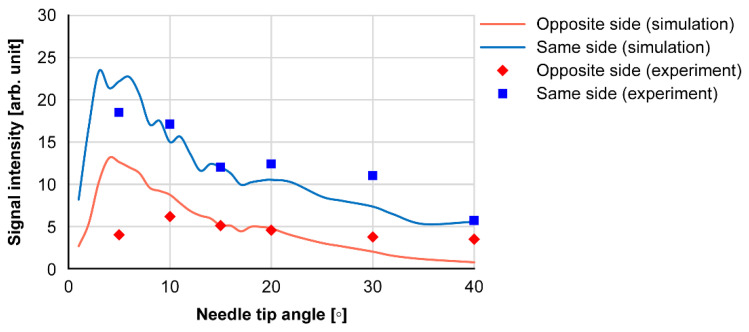
Needle tip angle and signal intensities received at the opposite and same sides.

**Table 1 diagnostics-12-00527-t001:** Potential applications and evidences.

Application	Evidence	References
Amniotic fluid puncture	Kratochwil	[2]
Aspiration of body fluids	Goldberg	[3]
Puncture of the liver and the pancreas	Holm	[4]
Puncture of the pancreas	Itoh	[5]
Puncture of the liver	Rasmussen	[6]
Biopsy of the kidney	Kristensen	[7]
Obtaining fluid or biopsy at amniocentesis, renal cyst aspiration, paracentesis, thoracentesis, kidney, liver masses, pancreas, and breast	Goldberg	[10]
Epidural puncture	Our idea	
Central venous catheter insertion	Our idea	

## Data Availability

Not applicable.

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
