# Peer review of "Needle Tip Detection Using Ultrasound Probe for Vertical Punctures: A Simulation and Experimental Study"

_diagnostics, 2022, doi:10.3390/diagnostics12020527_

Round 1

Reviewer 1 Report

adequate response

Author Response

Reviewer 1

  1. Reviewer comment: adequate response

Response: Thank you for your valuable comments.

Reviewer 2 Report

The authors describe the method of vertical puncture and claim, that the method is superior to other methods. 

Overall, there are several concerns with their paper: 

  • The introduction is way too long and should be condensed. In addition, the study hypothesis and aims of the study should be described more clearly. 
  • The methods section does not provide very much insight into what the authors did for the presented study and what they did in previous work. This has to be more focused. Many of the diagrams explaining the physics behind the method should be moved to a supplementary file. 
  • Results: Here, it is, again, not clear what the authors in the current study and what they had done previously. Also, ALL figures that are copyrighted by other organizations do, I assume, not have anything to do with the current study. Therefore, if they are created for this particular study, they should be removed entirely. 
  • Some claims of the authors are not supported by data. for example, the vertical puncture is more intuitive compared to other methods. This may be true but direct comparison is not presented here. Further, there are no sensitivity analyses presented that justify a "high sensitivity" (page 20, discussion section)
  • A table should be added summarizing potential applications and summarizing for which applications evidence already exists. 
  • all figures need an explanation of abbreviations. 
  • the overall number of figures needs to be reduced. suggest including 1-2 in the methods section and up to 6 for the results section. The rest should be either omitted or moved to a supplement. 
  • The discussion section mixes with the results section in many instances. Suggest to shorten and re-write the discussion keeping the following principles and structure in mind: summary of the main results, putting them into context with published data, discussion strengths and limitations, and providing conclusions at the end. 

Author Response

We agree with you and have incorporated your suggestions throughout the manuscript. We have answered all the comments from the Reviewer. Thank you for the suggestions. The following eight points have been corrected.

Reviewer 2

  1. Reviewer comment: The introduction is way too long and should be condensed. In addition, the study hypothesis and aims of the study should be described more clearly.

Response: The “Introduction” section has been rewritten to make it concise and to include the hypothesis and aims of the study (lines 39–117, pages 1–3).

  1. Reviewer comment: The methods section does not provide very much insight into what the authors did for the presented study and what they did in previous work. This has to be more focused. Many of the diagrams explaining the physics behind the method should be moved to a supplementary file.

Response: In the “Materials and Methods” section (lines 118–253, pages 3–6); we have described the experiments conducted in the previous studies as well as in this study. Further, the figures (Figures S1–S7 in lines 506–531, page 14) have been moved to the supplementary files.

  1. Reviewer comment: Results: Here, it is, again, not clear what the authors in the current study and what they had done previously. Also, ALL figures that are copyrighted by other organizations do, I assume, not have anything to do with the current study. Therefore, if they are created for this particular study, they should be removed entirely.

Response: We have described the results of the previous studies and those of the current study in the “Results” section (lines 255–352, pages 7–10). The unnecessary figures have been deleted and those required for comparison (Figure S8 in lines 533-536, page 14) have been moved to the supplementary files.

  1. Reviewer comment: Some claims of the authors are not supported by data. For example, the vertical puncture is more intuitive compared to other methods. This may be true but direct comparison is not presented here. Further, there are no sensitivity analyses presented that justify a "high sensitivity" (page 20, discussion section)

Response: The description about the intuitiveness and high sensitivity of vertical puncture has been deleted.

  1. Reviewer comment: A table should be added summarizing potential applications and summarizing for which applications evidence already exists.

Response: We have added a table presenting the potential applications and evidence (Table 1 in lines 476–477, page 13).

  1. Reviewer comment: all figures need an explanation of abbreviations.

Response: The abbreviations in the figure have been defined (Figure 1 in lines 56–60, page 2).

  1. Reviewer comment: the overall number of figures needs to be reduced. suggest including 1-2 in the methods section and up to 6 for the results section. The rest should be either omitted or moved to a supplement.

Response: We have included two figures in the “Materials and Methods” section and five figures in the “Results” section. We have moved the figures (Figure S9–S18 in the revised manuscript in lines 538–584, pages 14–16) required for illustration to the supplementary files.

  1. Reviewer comment: The discussion section mixes with the results section in many instances. Suggest to shorten and re-write the discussion keeping the following principles and structure in mind: summary of the main results, putting them into context with published data, discussion strengths and limitations, and providing conclusions at the end.

Response: The “Discussion” section has been rewritten (lines 354–483, pages 10–13). We have separated the experimental results and discussion; we have also incorporated the effects of the puncture spacer attachment, the advantages of vertical puncture, a discussion on the phase change at the needle tip, a discussion of the relationship between the needle tip angle and reception strength, the limitations of the study, potential applications and evidence and conclusions in the “Discussion” section.

Thank you for your valuable comments and queries and which gave us the opportunity to strengthen our manuscript. We have thoroughly checked the manuscript and incorporated the changes marked here. 

Round 2

Reviewer 2 Report

I have no further comments. Apart from being lengthy at times, the authors have improved their paper. 

This manuscript is a resubmission of an earlier submission. The following is a list of the peer review reports and author responses from that submission.

Round 1

Reviewer 1 Report

Authors have correctly responded to my comments.

Reviewer 2 Report

In this manuscript, there are many citations from the papers published in 2014. It is not enough for publication as a new study. The authors should pay more attention to the researches of the new investigation.